# Safety of Nonagenarians Receiving Therapeutic ERCP, Single Center Experience

**DOI:** 10.3390/jcm11175197

**Published:** 2022-09-02

**Authors:** Chia-Chang Chen, Wan-Tzu Lin, Chun-Fang Tung, Shou-Wu Lee, Chi-Sen Chang, Yen-Chun Peng

**Affiliations:** 1Division of Gastroenterology and Hepatology, Department of Internal Medicine, Taichung Veterans General Hospital, Taichung 40705, Taiwan; 2Department of Post-Baccalaureate Medicine, College of Medicine, National Chung Hsing University, Taichung 40227, Taiwan; 3School of Medicine, National Yang Ming Chiao Tung University, Taipei 11230, Taiwan; 4Department of Internal Medicine, Taichung Veterans General Hospital Chiayi Branch, Chiayi 60090, Taiwan

**Keywords:** endoscopic retrograde cholangiopancreatography, nonagenarians, post-endoscopic retrograde cholangiopancreatography pancreatitis

## Abstract

(1) Background: The complication rates for nonagenarians receiving therapeutic endoscopic retrograde cholangiopancreatography (ERCP) remain poorly understood. We aimed to determine whether nonagenarians were at an increased risk of ERCP-related complications. (2) Methods: We performed a retrospective study on therapeutic ERCP in nonagenarians from 2011 to 2016 at Taichung Veterans General Hospital. A control group comprising patients aged 65 to 89 years was used to compare demographic data and the outcomes of therapeutic ERCP with the nonagenarians. The risk factors for complications were determined by logistic regression model. (3) Results: There were 35 nonagenarians and 111 patients in the control group. Overall, complication rates were not statistically different between the two groups. However, advanced age was an independent predictor of complications in the multivariate analysis (odds ratio [OR] = 1.06; 95% confidence interval [CI] = 1.01–1.12; *p* = 0.049). End stage renal disease (ESRD) was another independent predictor of complications (OR = 4.87; 95% CI = 1.11–21.36; *p* = 0.036). Post-ERCP pancreatitis and bleeding were more common in ESRD patients than patients without ESRD. (4) Conclusions: Although nonagenarians receiving ERCP did not have more complications compared to elderly patients younger than 90 years, advanced age and comorbidity still affect the outcome of therapeutic ERCP in the elderly patients.

## 1. Introduction

After decades of development, endoscopic retrograde cholangiopancreatography (ERCP) has become an important examination and treatment modality for pancreato-biliary disease. Therapeutic ERCP requires advanced training in order to achieve mastery of this highly technical procedure [1,2]. ERCP, as well as therapeutic ERCP-related procedures, may result in complications, which include bleeding, pancreatitis, perforation, and cardiopulmonary distress [3]. ERCP procedure-related mortality has also been reported. 

Improvements in health care have extended the average life expectancy in many countries over the past few decades. Thus, as populations age, the diagnosis and treatment of age-related conditions and related complications are becoming increasingly important issues in geriatric care systems [4]. Therapeutic ERCP is a major treatment modality for pancreato-biliary diseases in elderly patients and its application in clinical care of the elderly is becoming more widespread. Age is considered an important factor for ERCP-related complications as well as prognosis [5,6].

The definition of elderly patients appears to vary considerably, particularly in the context of ERCP. The age range of elderly patients receiving ERCP tends to be from 65 to 80 years old [7,8,9,10,11]. Octogenarians comprise the largest subgroup of elderly patients receiving ERCP in the recent years. However, even older patients, such as those aged 90 years and older, are increasingly undergoing ERCP due to rising life expectancies. Previous studies on nonagenarians (individuals 90–99 years old) have investigated the outcomes of therapeutic ERCP and have found it to be effective and safe [12]. However, the very elderly seems to be at a higher risk of certain adverse events [13]. There was a controversy regarding the safety of ERCP for nonagenarian patients in the current literature [14,15]. It is uncertain whether ERCP may carry a higher risk for nonagenarians. There are also few data on the application of ERCP in the treatment of elderly patients in Taiwan. 

In the present study, we retrospectively reviewed data related to therapeutic ERCP for nonagenarian patients in our hospital. We aimed to determine whether nonagenarians were at a greater risk for ERCP adverse events compared with younger elderly patients.

## 2. Materials and Methods

### 2.1. Patients

This study is conducted at Taichung Veterans General Hospital, a tertiary medical center located in central Taiwan. It has an endoscopic unit which employs a standard protocol for therapeutic ERCP.

From January 2010 to December 2016, patients aged 90–99 years who underwent ERCP were recruited (nonagenarian group) and the index date of his or her first ERCP procedure was recorded. For each index date, consecutive patients aged 65 to 89 years old undergoing ERCP were selected as the control group.

Indications for therapeutic ERCP were common bile duct stone, cholangitis, or biliary drainage for obstructive jaundice. Before the procedure, written informed consent was obtained from all participating patients. In patients undergoing high-risk endoscopic procedures, such as endoscopic sphincterotomy (EST), we discontinue long-term anticoagulation or antiplatelet agents five to seven days before a high-risk procedure. All of the endoscopic or ERCP procedures were carried out in accordance with approved guidelines.

For the therapeutic ERCP, the endoscopic procedure was performed under conscious sedation with 0.5–5.0 mg intravenously administered midazolam and 25–50 mg pethidine. ERCP procedures were performed by experienced ERCP endoscopists. A trainee also participated in each of the procedures. Experienced endoscopists may supervise a trainee performing a specific procedure, such as duodenum intubation, locating the ampulla Vater, and cannulation, if the trainee is considered competent.

The ERCP procedures were performed with a standard endoscope (Olympus duodenoscope JF260V/TJF240/TJF260V). To access the biliary or pancreatic duct, we use ERCP catheter or sphincterotome with contrast injection method. Guidewire was inserted into desire duct after opacifying desire duct. Guidewire-assisted cannulation followed by precut may be used for cannulation if above method failed. Endoscopic sphincterotomy was performed if required. For bile duct stone removal, the standard techniques with basket or retraction balloon were performed. 

We collected demographic, American Society of Anesthesiologists (ASA) class, basic laboratory, indication of ERCP, concomitant diseases, biliary intervention, and all complications. This study was approved by the Ethics Committee of our Institutional Review Board (Taichung Veterans General Hospital IRB: CE -17014A).

### 2.2. Follow-Up of Patients after Procedure

Patients were monitored after ERCP. Their clinical symptoms, blood pressure, heart rate, and oxygen saturation were monitored for at least two hours. Then, patients were followed up until discharge from the hospital. Any patients with warning signs, such as abdominal pain or free air during procedure, received more intensive observation. Further evaluations, including imaging studies or laboratory data, were performed to determine whether there was perforation or pancreatitis if the clinical condition was deteriorated. If there were symptoms/signs of gastrointestinal bleeding after ERCP, endoscopic follow-up examination was conducted. 

### 2.3. Definition of Complications

Following complications were recorded in our study. The definition of bleeding was overt upper GI bleeding (e.g., melena, bloody vomiting), needing another endoscopic, or surgical intervention. Post-ERCP pancreatitis (PEP) is defined as a greater than three-fold elevation above the upper normal limit of post-procedure amylase or lipase combined with abdominal pain. Perforation was defined as evidence of free air during the procedure or on the follow-up imaging study. Cardiopulmonary distress was determined as deoxygenation (less than 90%), and patients could not recover after initial pulmonary resuscitation. Cholangitis was defined as new onset fever (>38.0 °C), jaundice, and liver biochemistry suggestive of biliary obstruction after ERCP.

### 2.4. Statistical Analysis

The statistical software SPSS (version 13.0; Chicago, IL, USA) was used for the statistical analysis. Categorical variables were compared by Chi-squared test, and continuous variables were expressed as median and IQR and compared by Mann–Whitney U test or Fisher’s Exact test. Logistic regression model was performed to analyze factors associated with complications, and significant factors (*p* < 0.05) in the univariate analysis were subjected to multivariate analysis to determine independent predictive factors. Statistical significance was defined as a *p* value of less than 0.05. 

## 3. Results

### 3.1. Characteristics of Patients

There were 35 nonagenarians and 111 patients aged 65–89 years receiving therapeutic ERCP during the study period. All patients’ age, gender, laboratory data, indications for ERCP, biliary intervention, concomitant diseases, ASA class, and adverse events are shown in Table 1. There were no significant differences in baseline laboratory data including bilirubin, platelet count, and prothrombin time international normalized ratio (INR) between the two groups. Therapeutic procedure such as EST, precut sphincterotomy, endoscopic retrograde biliary drainage, and lithotripsy were similar in the two groups. Associated pancreato-biliary diseases, including stone, malignancy, and benign biliary stricture, were also similar in the two groups. The ASA class was also similar between the two groups (Table 1).

### 3.2. Complications

The complications are shown in Table 1. There were fifteen patients (13.51%) in the control group and seven patients (20.00%) in the nonagenarian group who experienced any complications related to the ERCP procedure. Overall, the prevalence rates of complications were not statistically different between the two groups (Mann–Whitney U test, *p* = 0.506). The frequency of bleeding was not significantly different between the two groups (8.57% in nonagenarians, 1.80% in control group, *p*= 0.089). There was also no difference in the occurrence of PEP. Two nonagenarians (5.71%) had ERCP-related perforation (sphincterotomy-related perforation), and both recovered after supportive care. 

### 3.3. Risk Factors Related to Adverse Events in Elderly Patients 

The characteristics of the patients with complications were compared with those without complications (Table 2). The two subgroups were similar in gender, therapeutic procedures conducted, indications for ERCP, and concurrent pancreato-biliary disease. Patients who had complications were older (85 years versus 80 years, *p* = 0.023) and had a higher ASA class. The prevalence rate of end stage renal disease (ESRD) was higher in the complications group (22.73% versus 4.84%, *p* = 0.012). In the logistic regression model (Table 3), advanced age was an independent predictor of complications in the multivariate analysis (odds ratio [OR] = 1.06; 95% confidence interval [CI] = 1.00–1.12; *p* = 0.049). The ASA class was a significant predictor of complications only in the single variate model, but did not remain significant in the multivariate analysis (OR = 1.46; 95% CI= 0.51–4.23; *p* = 0.483). End stage renal disease (ESRD) was another independent predictor of complications (OR = 4.87; 95% CI = 1.11–21.36; *p* = 0.036). We further compared complications between the patients with ESRD with those without ESRD (Table 4). PEP was more common in ESRD patients than those without ESRD (36.36% versus 8.89%, *p* = 0.020). Post-ERCP bleeding was also more common in ESRD patients than those without ESRD (18.18% versus 2.22%, *p* = 0.046).

## 4. Discussion

We found the complication rate in patients receiving therapeutic ERCP was similar between the nonagenarian and the elder patients 65–89 years of age. Although nonagenarians receiving ERCP did not have more complications compared to elderly patients younger than 90 years, advanced age and comorbidity still affect the outcome of therapeutic ERCP in the elderly patients. Age and ESRD were independent risk factors for complications in the multivariate analysis. Among all patients in the study and control groups, the median age was 80 years old; the complication rate was low. (Perforation = 1.37%, bleeding that needs intervention = 3.42%, mortality rate = 0.68%, cholangitis = 2.05%, PEP = 10.96%.) 

The number of studies evaluating the safety of therapeutic ERCP in very old patients is still limited. Publications regarding the safety issue of super-elder people receiving ERCP in the recent 5 years are summarized in Table 5 [14,15,16,17,18,19,20,21]. These studies had variable designs, such as the definition of an elder group (ranged from 80–90 years old), control groups (elderly versus younger people or super-old versus elderly), and inclusion criteria (stones, therapeutic, diagnostic). Some studies found a higher complication rate [14]; others even found a higher mortality [16]. Some studies concluded that ERCP in elderly is safe and did not increase complication rates [19]. The conflicts encouraged more studies for this issue. Furthermore, the extreme age group (nonagenarian) is still a few in this list. Tomoya retrospectively evaluated the safety of endoscopic procedures for common bile duct stones in patients aged 85 years or older. In this retrospective study of 235 cases, 185 cases were aged younger than 85 years while 50 cases were aged 85 years or older (88 ± 3.0 years). No significant differences were found between the groups in terms of the incidence of complication after endoscopic procedures, including pancreatitis (16.8% [31/185] vs. 12.0% [6/50]), cholangitis (3.2% [6/185] vs. 2.0% [1/50]), and bleeding (1.1% [2/185] vs. 4.0% [2/50]). There were no risk factors for complications found in the study [18]. In another study conducted by Saito, 569 patients 75–89 years of age and 126 patients ≥ 90 years of age, who had native papilla and underwent therapeutic ERCP for choledocholithiasis, the rate of complete stone removal in patients 90 years of age was lower than that in patients 75–89 years. There was no significant difference in the incidence of post-ERCP complications between patients 75–89 years of age and those 90 years of age (7.7% vs. 9.5%, respectively; *p* = 0.47) [20]. Different from the study which only included patients who were hospitalized for CBD stones, our study included all therapeutic ERCP, including pancreato-biliary cancer (20.55%), common bile duct stones being only present in 67.12% of our patients. Similarly, there were no significant differences between the nonagenarian group and the 65–89-year-old group in terms of complication rates. However, there was an increase in the complications rate with increasing age in the whole cohort. Age may increase complication rates, which was observed in another study conducted by Takahashi including 137 patients aged ≥ 85 years who underwent therapeutic ERCP (indications including benign and malignant disease). In the study, Takahashi found that the group aged ≥ 90 years had fewer comorbid diseases but significantly increased adverse events compared with patients aged 85–89 years. In multivariate analysis, ≥90 years of age was a significant factor (*p* = 0.049) for adverse events [14]. The author pointed out the lack of comorbidities seen among the group aged ≥ 90 years suggests the possibility of selection bias. Patients aged ≥ 90 years with other underlying diseases might have been thought to be at too high a risk to undergo ERCP. Another limitation of Takahashi’s study is that the post-ERCP bleeding rate was not reported. Bleeding is an important adverse event for ERCP. The reason for the unreported bleeding rate is unknown. However, we think the EST rates were very low in Takahashi’s study (only 31%), which may cause very low or even no post-ERCP bleeding in their cohort. In our study, the EST rate was high (82.19%). The bleeding rate is also very low (3.42%). However, the bleeding rate was very high in those with ESRD (18.18%). Therefore, we need to carefully interpret safety data of nonagenarians undergoing therapeutic ERCP. There were too many factors (indications, comorbidities, procedures, and others) contributing to different types of adverse events (bleeding, pancreatitis, infection, and others); all these factors need to be included in future studies. The sample sizes of the nonagenarians included in past studies were still too small to put a robust conclusion.

Age may be a risk factor for ERCP complications because the elderly often has a higher Charlson comorbidity index and more comorbidities, such as cirrhosis, congestive heart failure, chronic obstructive pulmonary disease, chronic kidney disease, cerebrovascular accident, dementia, and anti-thrombotic medication. These factors tend to increase the risk of developing complications [6,16,20]. The procedure difficulty may also be greater for elderly patients. For example, duodenal diverticulum and duodenal deformity are seen more frequently in the elderly. The elderly often has larger common bile duct stones and a lower rate of stone removal [18]. Thus, nonagenarians require more ERCP procedures to eliminate bile duct stones [22]. In addition, the higher proportion of tumors in elderly patients, especially ampullary carcinoma, resulted in difficult cannulation [23], which was known as a factor for post-ERCP complications [21]. Age and multiple comorbidities are also associated with increased odds of inpatient mortality in patients undergoing ERCP [16]. In our study, the precut rate was used more often (18.49%) compared to a similar study. This may indicate that the cannulation was more difficult in our patients, and the patients were all old (≥65 years old). Pancreato-biliary malignancy was presented in 20.55% of patients. Our cohort represented the most common patient populations in referring ERCP centers. The patients are often old, fragile, complicated, or even had previous failed ERCP in other institutions. We can expect that factors associated with complications would be more common in this population and correlated with an increasing age, as presented in our study.

ESRD is another risk factor for complications in our cohort. We also found the rate of bleeding and PEP were significantly higher among the patients with ESRD. This result is the same as a previous large retrospective cohort study. ESRD and chronic renal disease were associated with higher post-ERCP adverse events including bleeding and PEP. The possible theory behind the increased association between ESRD and PEP may be papillary edema from fluid overload posing difficult biliary cannulation [24]. Many studies have shown an association between the overall rate of adverse events, especially PEP, and repeated initial cannulation attempts or the use of advanced techniques in difficult cases. In another study of 76 consecutive ERCPs for hemodialysis patients, the incidence of hemorrhage with EST was as high as 19%. The duration of hemodialysis was significantly longer in the patients with hemorrhage after EST than without. In our cohort, the bleeding rate for ESRD patients is 18.18%. EST may be needed to be avoided because of the high hemorrhage rate, particularly for patients with a long duration of hemodialysis [25]. This finding also reminded us that not only age, but also comorbidity are closely associated with ERCP complications.

Meanwhile, cardiopulmonary stress related to sedation has been shown to be an important concern in elderly patients. The rate of sedation-related adverse events was higher in older populations [26]. In our study, cardiopulmonary events related to sedation were very low (0.68%). In our institution, we use a very low dose of sedative agents (midazolam 0.5–5.0 mg and pethidine 25–50 mg), especially for much older patients. We also do not use propofol without the supervision of an anesthesiologist. Thus, we rarely need an anesthesiologist for our ERCPs. This may explain why we had a very low rate of cardiopulmonary events related to ERCP in our institution.

There were some limitations in this study. First, the number of cases was relatively small, and thus, the statistical power of the results was likely low. So, we were unable to analyze individual complications owing to the low rate of adverse events and small sample size. Second, as this was a retrospective study, not all of the relevant data could be included in the analysis. Detailed conditions in the ERCP procedure were not available. The number of patients in each group who did not undergo therapeutic ERCP despite the situation of its indication in the study period is unknown. There may be some selection bias. However, this hospital-based study on therapeutic ERCP in elderly patients could serve as the basis of further research, which could include more detailed data and long-term follow-up of our patients. Future prospective studies should also include more comprehensive data related to the ERCP procedure itself, such as procedure time, cannulation numbers, or prophylactic strategies for the prevention of adverse events in elderly patients receiving therapeutic ERCP.

## 5. Conclusions

We found that the incidence of post-ERCP complications in nonagenarians was similar to that in elderly patients 65–89 years of age. However, advanced age and comorbidity such as ESRD still affect the patient’s outcome in therapeutic ERCP. Endoscopists should know that not only age, but also a patient’s conditions are important for performing therapeutic ERCP in elderly patients. Further large studies including more patients’ condition and procedure details are needed to clarify the result of our study.

## Figures and Tables

**Table 1 jcm-11-05197-t001:** Summary of patient characteristics in the nonagenarians and control group (aged 65–89 years).

	Total (*n* = 146)		Age > 90				*p* Value
			No (*n* = 111)		Yes (*n* = 35)		
Age	80	(73–88.25)	77	(71–82)	91	(90–93)	<0.001 **
Male	102	(69.86%)	78	(70.27%)	24	(68.57%)	1.000
Total bilirubin (mg/dL)	1.3	(0.6–3.73)	1.3	(0.6–3.7)	1.4	(0.6–4.8)	0.745
INR	1.04	(0.98–1.09)	1.04	(0.99–1.09)	1.03	(0.98–1.1)	0.956
Platelet (10^9^/L)	191	(143.75–249)	191	(144–249)	192	(130–239)	0.541
ASA class 3–4	57	(39.04%)	39	(35.14%)	18	(51.43%)	0.085
ESRD ^f^	11	(7.53%)	8	(7.21%)	3	(8.57%)	0.725
Pancreatitis ^f^	20	(13.70%)	16	(14.41%)	4	(11.43%)	0.783
CBD stone	98	(67.12%)	73	(65.77%)	25	(71.43%)	0.678
Benign stricture ^f^	16	(10.96%)	11	(9.91%)	5	(14.29%)	0.536
Perihilar cancer ^f^	11	(7.53%)	8	(7.21%)	3	(8.57%)	0.725
Periampulla vater cancer	21	(14.38%)	18	(16.22%)	3	(8.57%)	0.397
Cancer	30	(20.55%)	25	(22.52%)	5	(14.29%)	0.417
EST	120	(82.19%)	90	(81.08%)	30	(85.71%)	0.710
Precut	27	(18.49%)	22	(19.82%)	5	(14.29%)	0.627
ERBD	48	(32.88%)	41	(36.94%)	7	(20.00%)	0.098
Lithotripsy ^f^	9	(6.16%)	7	(6.31%)	2	(5.71%)	1.000
Complications							
Bleeding ^f^	5	(3.42%)	2	(1.80%)	3	(8.57%)	0.089
PEP ^f^	16	(10.96%)	11	(9.91%)	5	(14.29%)	0.536
Perforation ^f^	2	(1.37%)	0	(0%)	2	(5.71%)	0.056
Cardiopulmonary distress ^f^	1	(0.68%)	1	(0.90%)	0	(0%)	1.000
Mortality ^f^	1	(0.68%)	0	(0%)	1	(2.86%)	0.240
Cholangitis ^f^	3	(2.05%)	2	(1.80%)	1	(2.86%)	0.563
Any complications	22	(15.07%)	15	(13.51%)	7	(20.00%)	0.506

Mann–Whitney U test. Chi-Square test. ^f^ Fisher’s Exact test. ** *p* < 0.01. Continuous data were expressed median and IQR. Categorical data were expressed number and percentage. ASA: American Society of Anesthesiologists; INR: international normalized ratio; ESRD: end stage renal disease; CBD: common bile duct; EST: endoscopic sphincterotomy; ERBD: endoscopic retrograde biliary drainage; PEP: post-endoscopic retrograde cholangiopancreatography pancreatitis.

**Table 2 jcm-11-05197-t002:** Patients who suffered from complications versus those who did not.

	Complications				*p* Value
	No (*n* = 124)		Yes (*n* = 22)		
Nonagenarians	28	(22.58%)	7	(31.82%)	0.506
Age	80	(72–87)	85	(79–91)	0.023 *
Male	87	(70.16%)	15	(68.18%)	1.000
Total bilirubin > 2 mg/dl	50	(40.32%)	8	(36.36%)	0.910
INR > 1.15 ^f^	14	(11.29%)	4	(18.18%)	0.478
Platelet < 150(10^9^/L)	37	(29.84%)	6	(27.27%)	1.000
ASA class 3–4	44	(35.48%)	13	(59.09%)	0.036 *
ESRD ^f^	6	(4.84%)	5	(22.73%)	0.012 *
Pancreatitis ^f^	19	(15.32%)	1	(4.55%)	0.311
CBD stone	83	(66.94%)	15	(68.18%)	1.000
Benign stricture ^f^	15	(12.10%)	1	(4.55%)	0.467
Perihilar cancer ^f^	10	(8.06%)	1	(4.55%)	1.000
Periampulla vater cancer	18	(14.52%)	3	(13.64%)	1.000
Cancer	25	(20.16%)	5	(22.73%)	0.778
EST	103	(83.06%)	17	(77.27%)	0.547
Precut	24	(19.35%)	3	(13.64%)	0.766
ERBD	41	(33.06%)	7	(31.82%)	1.000
Lithotripsy ^f^	8	(6.45%)	1	(4.55%)	1.000

Chi-Sqaure test. ^f^ Fisher’s Exact test. * *p* < 0.05. ASA: American Society of Anesthesiologists; INR: international normalized ratio; ESRD: end stage renal disease; CBD: common bile duct; EST: endoscopic sphincterotomy; ERBD: endoscopic retrograde biliary drainage.

**Table 3 jcm-11-05197-t003:** Risk factors of complications for whole cohort in logistic regression model.

	Simple Model			Multiple Model		
	OR	95%CI	*p* Value	OR	95%CI	*p* Value
Nonagenarians	1.60	(0.59–4.31)	0.353			
Age	1.06	(1.01–1.12)	0.031 *	1.06	(1.00–1.12)	0.049 *
ASA class 3–4	2.63	(1.04–6.63)	0.041 *	1.46	(0.51–4.23)	0.483
ESRD	5.78	(1.59–21.04)	0.008 **	4.87	(1.11–21.36)	0.036 *

Logistic regression. * *p* < 0.05, ** *p* < 0.01. ASA: American Society of Anesthesiologists; ESRD: end stage renal disease.

**Table 4 jcm-11-05197-t004:** Complications between patients with ESRD and patients without ESRD.

	No (*n* = 135)		Yes (*n* = 11)		*p* Value
PEP ^f^	12	(8.89%)	4	(36.36%)	0.020 *
Bleeding ^f^	3	(2.22%)	2	(18.18%)	0.046 *
Cholangitis ^f^	3	(2.22%)	0	(0%)	1.000
Perforation ^f^	1	(0.74%)	1	(9.09%)	0.145
Cardiopulmonary distress ^f^	1	(0.74%)	0	(0%)	1.000
Complications ^f^	17	(12.59%)	5	(45.45%)	0.012 *
Mortality ^f^	1	(0.74%)	0	(0%)	1.000

Mann–Whitney U test. Chi-Square test. ^f^ Fisher’s Exact test. * *p* < 0.05. Continuous data were expressed mean ± SD. Categorical data were expressed number and percentage. PEP: post-endoscopic retrograde cholangiopancreatography pancreatitis.

**Table 5 jcm-11-05197-t005:** Publications regarding safety issue of elder people receiving ERCP in recent 5 years.

Author Year	ERCP Indication	No. of Elder	Age of Elder	Age of Control	Increase Complication Rate in Elder	Risk Factor for Complications
Sobani ZA et al., 2018 [16]	Diagnostic Therapeutic	74	>90	18–89	Yes ^A^	Charlson Comorbidity Index ≥ 2 Emergency procedures
TakahashiK et al., 2018 [14]	Therapeutic	25	>90	85–89	Yes	Age > 90
GaleazziM et al., 2018 [17]	Diagnostic Therapeutic	50	>80	65–79	No	Not found
Iida T et al., 2018 [18]	Stone	235	>85	< 85	No	Not found ^B^
Yang JH et al., 2018 [19]	Therapeutic	141	>80	<65	No	Not found
Saito H et al., 2019 [20]	Stone	126	>90	75–89	No	Not found ^B^
Tabak F et al., 2020 [21]	Therapeutic	146	>80	<80	No	Charlson Comorbidity Index ≥ 2 Difficult cannulation
Ogiwara S et al., 2020 [15]	Therapeutic	66	>90	70–79	No	Not found
This study	Therapeutic	35	>90	65–89	Yes	Age ESRD

^A^ Inpatient mortality. ^B^ The complete removal rate of bile duct stones was lower in elder group. ERCP: endoscopic retrograde cholangiopancreatography; ESRD: End stage renal disease.

## Data Availability

Data are available on request from the authors.

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
