# Peer review of "Safety of Nonagenarians Receiving Therapeutic ERCP, Single Center Experience"

_jcm, 2022, doi:10.3390/jcm11175197_

Round 1

Reviewer 1 Report

Cealing of care for nonagenarians presenting with acute biliary pathology is a common gastroenterological/surgical dilemma. This study has clearly demonstrated that risk is not higher comparing to younger age group. However, the risk is higher for patients with ESRD. I would like to congratulate authors, as this study will help when counseling patients for ERCP and having discussion about risks. Clinically very relevant. I would add ASA grades for comparison of two groups. 

Reviewer 2 Report

This manuscript is an original article that retrospectively investigated the safety of therapeutic ERCP in nonagenarians, by comparing nonagenarians with patients aged 65 to 89 years. The authors demonstrated that complication rates were not statistically different between two groups, and that advanced age and end stage renal disease was independent predictors of complications.

This study was conducted well, and presented clearly. And, it contains informative information, which will be of interest to researchers and clinicians in the field.

However, I have serious concern in this manuscript. Several similar previous studies have already published and shown the safety of ERCP-related procedure in elderly patients with larger population. Furthermore, information regarding performance status, comorbidities, administration of antithrombotic medication, detailed condition in ERCP procedure (procedure time, canulation time, pancreatic duct canulation, and so on) should be described as those factors can influence outcomes. The number of patients in each group who did not underwent therapeutic ERCP despite the situation of its indication in the study period should be provided in order to reduce selection bias.

Minor

1. Please explain the situation of two cases who suffered from perforation in detail.

2. Please confirm which sedative drug was used, diazepam or midazolam. The used drug differs between in Method and in Discussion.

Reviewer 3 Report

Well written study . It provides reassurance that ERCP procedure when indicated is necessarily needed even in very elderly patients. There are few risk factors that every endoscopist should take into a consideration though. 

I feel that there could be a reference to the types of comorbitites that the patient aged over 90 presented with .otherwise , it is an informative study despite its  retrospective nature. 

Reviewer 4 Report

There has been limited number of publications related to the subject of the safety of ERCP in advanced age groups. This is due, that such a vast subpopulation of elderly patients is present only in some countries.  In western countries collecting a group of 35 patients older than 90 would be a long and rather impossible venture if collected by one center. In my personal experience, a patient (nonagenarian)  selected for ERCP comes once a year.

Approving an elderly patient for ERCP is the case-by-case selection process.  From the general experience, it is known and expected that such a group of patients carry a higher risk of complications. Because qualifying for  ERCP in this group of patients is a highly selective process the level of complications may not be fully elucidated.  The final outcome relies more on the general patient condition, rather than on a procedure (in this case ERCP). It is also known, for example in HPB surgery that liver resection is safe in the elderly population as in younger patients. But risks of an unsuccessful operation lie in the patient's general condition which is determined by the severity of comorbidities.  From this perspective, the manuscript brings no new insights. Even in a situation of a higher rate of postoperative complications the treatment strategy would not be changed. 

In the presented group nonagenarians' average age was 91 ranging from 90 to 93. It seems to stay rather on the younger side.  The question arises if this is not a selection bias. Or maybe, there was not a single patient older than 93.

Of course, in the case of ERCP selection bias  as well as strict selection criteria were present.  A patient must be fit enough to have a safe ERCP. On the other hand, in that group of patients, there are limited options for other treatments.  Therefore ERCP remains the only treatment option even taking into account an increased risk of complications.

It might be suspected that ERCP procedure “per se” is safe, carrying the same number of complications in the nonagenarian and control groups because the procedure is carried out in the same mode in both groups and the risks are generally the same.

It might also be anticipated that the majority of possible complications will be associated with age and related comorbidities.   Authors showed that complications rate even in the control group tents to stay higher over age 85 (22.73%). The complication rate in nonagenarians appears to be acceptable. Table 1 shows 7 patients which are 20.00% of any complications. It appears that 7 patients had 12 various complications. It must have been a group with more than one complication.   

ESRD is serious comorbidity in elderly patients independently of other chronic conditions. Only a small number of all patients have been diagnosed ESRD (11 in both groups, only 3 in the nonagenarians subgroup). I would argue that there is not enough data to support the suggestion that ESRD is an independent predictor of complications in nonagenarians.  Primary because the group is simply too small. It would be interesting to know what was the mean age of patients with ESRD in the control group. To me, ESRD presents the general condition of a patient and it should be taken into account in selecting any patients, not only in the age group over 90 years. I believe that a group is too limited to draw such a strong conclusion about ESRD as an independent predictor of complications in nonagenarians unless the results presented in Table 3 relate to all 146 patients.  There have been previous publications related to chronic kidney failure and complication rates after ERCP. From this perspective, this is yet another paper confirming the known relationship.    

It is obvious,  that it has been done proper statistical work but the tables seem to me unclear. As it was too much data put into each table. In my opinion Table, 4 adds nothing new to the manuscript and has no obvious connections with the manuscript title which is: Safety of ERCP in nonagenarians. Table 4 is about ESRD which is a known factor affecting complication rate and has no impact on explaining the purpose of the manuscript.

In the discussion section, the authors state that nonagenarian is not a risk factor for complications.  By definition, a nonagenarian is a person in the age range between 90 and 99. One can argue that the authors already drew the conclusion that age is the risk factor as is stated as the advanced age in the abstract. To support the thesis authors showed complication rates in a subpopulation over 85 years. What made such a conclusion is related to selection and inclusion criteria. (For example, changing the inclusion criteria to the younger population aged 85-89 probably would change the result of the study. I would make this statement clear or rephrase it or discuss it a bit further. In my opinion, nonagenarian equals advanced age. And advanced age is a risk factor.

Round 2

Reviewer 2 Report

The revised manuscript is much improved. However, the following minor issues should be addressed.

Minor

1. (P2L33-35) This sentence sounds strange. It may contain a grammatical error.

2. I recommend that the authors introduce a controversy regarding the safety of ERCP in for nonagenarian patients using some literatures.

3. A table you provided in your response (Publications regarding safety issue of elder people receiving ERCP in recent 5 years) can help readers’ understanding. I recommend that the authors provide the table in the main text, adding more detailed information (authors, rate of therapeutic ERCP, success rate, complication rate, etc.), and expand discussion based on those data.
